# Cannabis consumption is associated with altered steroid metabolism in young men

Mathieu Galmiche [1,2,3], Isabel Meister [1,2,3], Fanny Zufferey[3,4], Michel F. Rossier [3,4,5], Rita Rahban [3,6], Alfred Senn[3,6], Serge Nef[3,6], Julien Boccard [1,2,3] & Serge Rudaz [1,2,3] ✉

## Abstract

**Background** Cannabis use has been hypothesized to alter endocrine function. We aimed at investigating this hypothesis through extended steroid profiling in young men.
**Methods** Using liquid chromatography – tandem mass spectrometry (LC-MS/MS), 70 endogenous steroids were reliably identified in serum samples from 47 cannabis consumers and 47 controls. Seven major steroids were subject to absolute quantification, while the others were considered as relative concentrations.
**Results** This cross-sectional study shows a global increase in androgen levels among cannabis consumers. Androstenedione (A4), testosterone (T), and dihydrotestosterone (DHT) are among the most significantly increased steroids. In contrast, C11-oxy androgens show no significant change upon cannabis use. This pattern suggests that phytocannabinoids might selectively affect gonadal androgen synthesis without altering adrenal or peripheral pathways, possibly via direct effects on the testes, or disruption of the hypothalamic–pituitary–gonadal (HPG) axis function. Additionally, two progesterone metabolites, 11β-hydroxyprogesterone (11β-OHP4) and 5β-dihydroprogesterone (5β-DHP4), are markedly elevated in cannabis consumers. When the cannabis user group is stratified according to the corresponding usage biomarkers, 11β-OHP4 proves to be a biomarker of general exposure, whereas 5β-DHP4 displays a dose-dependent relationship.
**Conclusions** These findings highlight the value of extended steroid profiling for investigating hormonal variations and evidence a possible link between cannabis consumption and altered male endocrine function.

## Plain language summary

Up to now, the effect of cannabis use on male hormones has been unclear, with previous studies having differing findings. We compared circulating concentrations of several hundred hormones, including androgens, in a cohort of young Swiss men differing in cannabis use status. Cannabis users showed higher levels of bioactive testicular androgens, which play a critical role in male reproductive function. However, levels of androgens produced in the adrenal gland were not altered. These results suggest that cannabis use in young men perturbs testicular hormone synthesis.

According to the World Health Organization, 2.5% of the world population (i.e., about 150 million people) would consume cannabis (also called marijuana)[1]. In parallel with increasing rates of recreational use, cannabis is now widely legalized or authorized for medical purposes, making the assessment of its potential adverse effects a growing public health priority. The impact of cannabis on the male reproductive system has been particularly documented but remains controversial[2–5]. Several studies reported altered semen parameters, such as a lower sperm count, concentration, motility, and viability in semen samples from men exposed to phytocannabinoids[2,4,6]. These adverse effects are thought to be mediated by the endocannabinoid system (ECS), which comprises lipid-derived neurotransmitters (endocannabinoids) and their receptors (CB1 and CB2), which

are expressed throughout the hypothalamic–pituitary–gonadal (HPG) axis[5,7]. Δ⁹-tetrahydrocannabinol (THC), the primary psychoactive compound in cannabis, can bind to these two receptors[4,5]. This interference of THC with the endocannabinoids in the ECS could affect the homeostasis of the HPG axis, which is essential for male reproductive function. Specifically, it could impair the regulation of key reproductive hormones in the hypothalamus and anterior pituitary, including the gonadotropin-releasing hormone (GnRH), the luteinizing hormone (LH), the follicle-stimulating hormone (FSH), and the sex hormones, such as androgens and estrogens[4–6,8,9]. As CB1 and CB2 receptors are also present in the testes, specifically in Leydig cells, phytocannabinoids may also directly alter testicular steroidogenesis[5].

[1]School of Pharmaceutical Sciences, University of Geneva, Geneva, Switzerland. [2]Institute of Pharmaceutical Sciences of Western Switzerland, University of Geneva, Geneva, Switzerland. [3]Swiss Centre for Applied Human Toxicology (SCAHT), Basel, Switzerland. [4]Service of Clinical Chemistry and Toxicology, Central Institute of Hospitals, Hospital of Valais, Sion, Switzerland. [5]Department of Internal Medicine, Faculty of Medicine, University of Geneva, Geneva, Switzerland. [6]Department of Genetic Medicine and Development, University of Geneva, Geneva, Switzerland. ✉e-mail: serge.rudaz@unige.ch

The current state of the art on the hormonal effect of cannabis consumption in males, especially its impact on testosterone (T) levels, has frequently been described as "inconsistent", "contradictory", or "conflicting"[6,10,11]. The only article reporting a significant decrease in circulating T levels in men consuming cannabis dates back to 1974, involved a small cohort (20 consumers compared to 20 controls), and lacked control for key confounding factors[12]. In contrast, numerous subsequent studies have found no significant differences in blood testosterone concentrations between cannabis users compared to non-users[2,10,13–16]. However, these studies were either limited by small sample sizes[13], or focused on subfertile men in clinical settings, which are not representative of the general population[14–16].

More robust cross-sectional studies conducted in recent years in Denmark and the United States have evaluated thousands of young men from the general population and consistently reported higher serum T levels in cannabis users compared to non-users[11,17–19]. These findings, drawn from large, well-characterized cohorts, suggest a positive association between cannabis use and testosterone, which could be more pronounced either with frequency of use[17] or with recency of regular use[11]. Despite these advances, existing studies are limited by several factors: reliance on self-reported cannabis use without biomarker confirmation, difficulty accounting for all potential confounders, and a narrow focus on testosterone without comprehensive hormonal profiling.

In Switzerland, the mandatory military enrollment for men between 18 and 22 years offers the possibility to recruit participants from the general population and evaluate their reproductive health[20]. Based on this vast recruitment campaign, sub-cohorts were created to investigate potential associations between male reproductive health outcomes and specific environmental factors. In particular, the gonadotropin axis function was studied in hundreds of participants and associated with their cannabis consumption status, which was confirmed by analyses of phytocannabinoid levels in biological fluids. This work concluded in increased concentrations of endocannabinoids, androgens, estradiol, and sex hormone binding globulin in cannabis smokers, with higher significance in chronic and recent consumers[21].

Gaining further insights into steroid metabolism in young men of reproductive age is essential to assess the potential hormonal imbalances associated with cannabis use. Most of the current state of the art is restricted to serum testosterone, with few exceptions, such as a recent Swiss study that also monitored androstenedione (A4), cortisol (F), and dehydroepiandrosterone sulfate (DHEAS)[21].

To address this limitation, the present investigation applies an extended steroid profiling approach to a subset of the Swiss cohorts described previously[20,21], comprising 47 cannabis users and 47 matched controls. This extended profiling performed with liquid chromatography (LC) hyphenated to tandem mass spectrometry (MS/MS) covers 171 target steroids from diverse subclasses, including androgens, progestogens, estrogens, corticosteroids, bile acids, oxysterols, and phase II metabolites (glucuronides and sulfates). Among the 70 reliably detected endogenous steroids, alterations in metabolic patterns related to testicular steroidogenesis are associated with cannabis consumption, providing insight into its systemic endocrine effects.

## Methods

### Sample selection
Participants were recruited nationwide in Switzerland from 2005 to 2017 as previously described[20]. The ethics committees of the cantons of Vaud (17-01-2005, 01/02), Zürich (EK-StV-Nr. 27-2006), Ticino (Rif.CE 1886), and Geneva (2016-01674) approved the present study. All participants signed a written informed consent. Serum samples were retrieved from the study of Zufferey et al.[21]. Zufferey et al. quantified cannabis biomarkers (THC and THC-COOH) in the same serum samples that were studied in the present work (see Section S10)[21]. Two groups were constructed to differentiate 47 participants with confirmed cannabis consumption (i.e., declared consumption and positive concentrations of THC and THC-COOH detected in serum) from 47 participants with no detectable THC and THC-COOH in

serum and who did not declare any cannabis use (categorized as controls). All participants were aged 18–23 at the time of sampling. Most samples were taken in late afternoon, and there was no significant difference in sampling time of day between THC-positive and control groups, considerably reducing the influence of diurnal variations as a confounder. BMI, which might also be a confounding factor, was available for each participant. The group of cannabis consumers was further separated into chronic or occasional consumers according to their circulating THC-COOH level[21,22]. Participants were considered chronic users when THC-COOH was quantified at more than 40 μg/L in serum. All cannabis users were defined according to their declared substance use within the previous 7 days. Samples from these participants reflect very recent cannabis use (<12 h, based on the model of Huestis et al.[23]).

### Chemicals
LC-MS "Optima" grade solvents, i.e., Acetonitrile (ACN), Methanol (MeOH), and Water ($H_2O$), were purchased from Fisher Scientific. Formic Acid (FA) was acquired from Biosolve Chimie at ULC-MS purity (>99%). Ammonium Fluoride ($NH_4F$) (>99.99% purity) was purchased from Sigma-Aldrich (Merck KGaA).

Analytical standards of endogenous steroids and $^{13}C$-labeled internal standards were supplied by Sigma-Aldrich (Merck KGaA), Steraloids Inc., and LGC Standards (LGC Ltd).

### Sample preparation
Between collection and extraction, all serum samples were stored at −80 °C. The procedure for the preparation of serum samples for multi-targeted steroid analysis was described in detail elsewhere[24]. Briefly, 750 μL of protein precipitation solution (ACN / MeOH, 9:1 v/v) containing $^{13}C$-labeled internal standards was added to 250 μL serum samples. After centrifugation, supernatants were filtered through HLB Prime 30 mg cartridges (96-well format, Waters Corp.). Extracts resulting from this "reversed solid-phase extraction" were evaporated to dryness and reconstituted in 50 μL Water / Methanol (1:1, v/v). The injection volume for each sample at the LC-MS/MS analysis was 5 μL.

### LC-MS/MS analysis
The multi-targeted LC-MS/MS method for extended steroid profiling was previously described in detail[24]. According to this protocol, seven steroid metabolites were subject to absolute quantification through a one-point internal calibration strategy, thanks to the commercial availability of 13C-labeled standards. The other steroids were analyzed semi-quantitatively using MRM peak areas. This corresponds to a semi-targeted assay according to the classification of Beger et al.[25].

The separation of steroids was performed with a Biphenyl stationary phase (Restek Raptor Inert Biphenyl, 2.1 × 100 mm, 1.8 μm) and a $H_2O$/MeOH mobile phase gradient (from 40 to 100 % methanol in 18 min). LC flow rate was 0.4 mL/min. A concentration of 0.01 % of FA was added to the mobile phase. $NH_4F$ was added post-column to enhance the ionization of steroids.

Mass spectrometry was achieved with a Xevo TQ-XS Triple Quadrupole equipped with a ZSpray ESI source (Waters Corp.). Multiple Reaction Monitoring (MRM) mode was used for data acquisition, utilizing MS/MS transitions that were previously optimized on neat standards. Both negative polarity and positive polarity transitions could be acquired simultaneously (polarity switching). The selected transitions for the cohort acquisition stemmed from a preliminary analysis of a pooled QC sample from this cohort, which suggested the tentative detection of 94 endogenous steroids out of the 171 target compounds, along with the 14 isotope-labeled internal standards (see Supplementary Data 1).

### Data processing
MRM chromatograms were acquired in MassLynx (Version 4.2, Waters Corp.) and processed in Skyline (Version 24.1, "molecule" interface, MacCoss Lab Software) with manual peak verification and integration.

**Table 1 | Absolute concentrations (in nmol/L) of seven major steroids in serum samples from cannabis users and non-users. _P_-values were obtained from two-tailed t-tests without multiple comparison adjustment**

| | THC-positive ($n = 47$) | | | Controls ($n = 47$) | | | Difference between groups (THC + - THC-) | |
|---|---|---|---|---|---|---|---|---|
| | Min | Max | Mean | Min | Max | Mean | Difference | _p_-value |
| A4 | 1.64 | 7.19 | 3.70 | 1.12 | 5.96 | 2.96 | +0.75 | 0.008 (**) |
| T | 7.2 | 32.2 | 19.3 | 6.6 | 28.9 | 15.7 | +3.5 | 0.002 (**) |
| 17α-OHP4 | 0.46 | 3.93 | 1.96 | 0.42 | 3.77 | 1.62 | +0.34 | 0.03 (*) |
| P4 | 0.169 | 0.594 | 0.297 | 0.143 | 0.513 | 0.272 | +0.025 | 0.19 (ns) |
| S | 0.09 | 3.45 | 1.07 | 0.16 | 3.06 | 1.01 | +0.06 | 0.70 (ns) |
| F | 56 | 366 | 235 | 60 | 413 | 241 | −6 | 0.72 (ns) |
| E | 14.7 | 58.4 | 33.0 | 11.4 | 51.1 | 33.5 | −0.5 | 0.79 (ns) |

Peak areas of endogenous steroids were normalized by peak areas of spiked [13]C-labeled internal standards (SILs) in the same sample. The attribution of a given SIL to a given endogenous compound was made based on the following criteria, by decreasing order of importance: mass spectrometry acquisition polarity, steroid class, and retention time difference. A summary of the SIL/analyte pairs is presented in Supplementary Data 2.

Steroid features were excluded from the dataset if they were absent from at least 50% of the samples from this study. A peak was considered missing in a sample if its peak area was lower than the mean peak area measured in the procedural blanks. Missing values were replaced by one-third of this mean "blank" peak area before multivariate analysis. Compounds were also excluded if the coefficient of variation (CV) of their normalized peak area in 10 pooled QCs exceeded 30%. A summary of quality control parameters for all the steroid compounds is given in Supplementary Data 3.

The determination of absolute concentrations of seven steroids for which SILs were commercially available was performed using a previously described one-point calibration strategy and an automated in-house workflow implemented in Python 3.9.[24,26].

## Statistics

The design of this clinical study involved two groups of 47 cannabis users and 47 matched controls. Among the cannabis users, 14 were classified as "chronic" while 33 were classified as "occasional" for the corresponding comparisons.

Univariate analyses were conducted with Prism (Version 10.3.1, GraphPad). T-tests were performed with or without Welch correction, depending on the homogeneity of variance in the two groups. Two-tailed _p_-values were calculated. Welch correction was applied when the variance in the two groups was significantly different (F-test, $p < 0.05$).

Multivariate analyses, including Principal Component Analysis (PCA), Orthogonal Partial Least Square—Discriminant Analysis (OPLS-DA), and Partial Least Squares regression (PLS), were performed after unit variance scaling using the software SIMCA (Version 17.0.2, Sartorius AG).

## Results & Discussion

### The concentration of bioactive androgens is significantly higher in cannabis users

Using the previously proposed one-point internal calibration approach[24], seven major compounds of the steroid biosynthesis pathway in humans were accurately quantified using their corresponding [13]C-labeled standard: androstenedione (A4), testosterone (T), 17α-hydroxyprogesterone (17α-OHP4), progesterone (P4), 11-deoxycortisol (S), cortisol (F), and cortisone (E) (Table 1). The absolute concentrations measured were consistent with established reference ranges for men in their early twenties[27], and aligned well with previous findings in a similar Swiss cohort[21].

Notably, serum levels of A4 and T, the two quantified androgens, were significantly higher in THC-positive individuals ($p = 0.008$ and $p = 0.002$, respectively). A modest but statistically significant increase was also observed for 17α-OHP4 ($p = 0.03$). In contrast, no significant differences were observed for P4, S, F, and E.

This finding supports the conclusion drawn by Gundersen et al. in a large cross-sectional cohort from Denmark, which evidenced a 7% higher T concentration in marijuana users after adjustment for confounders[17]. The same observation was reported in another large cross-sectional study led in the U.S., where T was measured with higher concentrations in THC-positive participants, regardless of the frequency and recency of use[19].

Although displaying a smaller androgenic activity than T, A4 is the major precursor of T and is thus essential in masculine sex hormone metabolism[28]. Its significantly higher concentration in cannabis consumers further supports previous findings of Zufferey et al.[21].

Absolute quantitative data on these seven major steroids were complemented by extended steroid profiling. Out of the 171 target steroids (Supplementary Data 1), 77 were consistently identified in serum, with 70 meeting stringent quality control criteria (see Supplementary Fig. S1, Supplementary Data 3). These encompassed a broad range of steroid subclasses: 17 androgens, 15 progestogens, 3 estrogens, 20 corticosteroids, 3 oxysterols, 9 glucuronides, 5 sulfates, and 5 bile acids (see Supplementary Fig. S2). This depth of profiling exceeds that of most previous targeted or untargeted steroidomic studies in human serum[29–32], and closely mirrors the steroidome found in certified human blood reference materials[24]. When no absolute concentration was determined, data analysis of the extended steroid profile was achieved using analyte peak areas normalized by peak areas of 13C-labeled internal standards (ISTDs) as representative of concentration.

Within the extended steroid profile, 5α-dihydrotestosterone (DHT) was of particular interest given the previous observation on T and A4. DHT is the most potent androgen in humans, even more than T, as it features a twice higher ability to bind to the androgen receptor, and a five-fold lower dissociation rate[33]. The present study demonstrates that serum DHT is significantly higher in THC-positive men ($p = 0.029$). This confirms the interest of measuring DHT and other androgens in addition to T to support hypotheses regarding steroidogenesis and androgen activity in men. As summarized in Fig. 1, the levels of all three bioactive androgens of gonadal origin were significantly higher among cannabis users.

### Adrenal androgen synthesis is not affected by cannabis consumption

Further insights into the complex male steroidome in the context of cannabis usage were obtained using multivariate analysis. An Orthogonal Partial Least Squares—Discriminant Analysis (OPLS-DA) model was used to discriminate THC-positive ($n = 47$) and THC-negative ($n = 47$) participants based on their circulating steroid profile (M1, see Section S2 of the Supplement). A 7-fold cross-validation of the optimal model with one predictive and one orthogonal component resulted in the following metrics: $R^2Xp = 0.055$, $R^2Y = 0.49$, $Q^2 = 0.19$. Although the low value of $Q^2$ indicates a low predictive ability, all $Q^2$ values obtained under the null hypothesis in a permutation test with 100 permutations were markedly lower than the

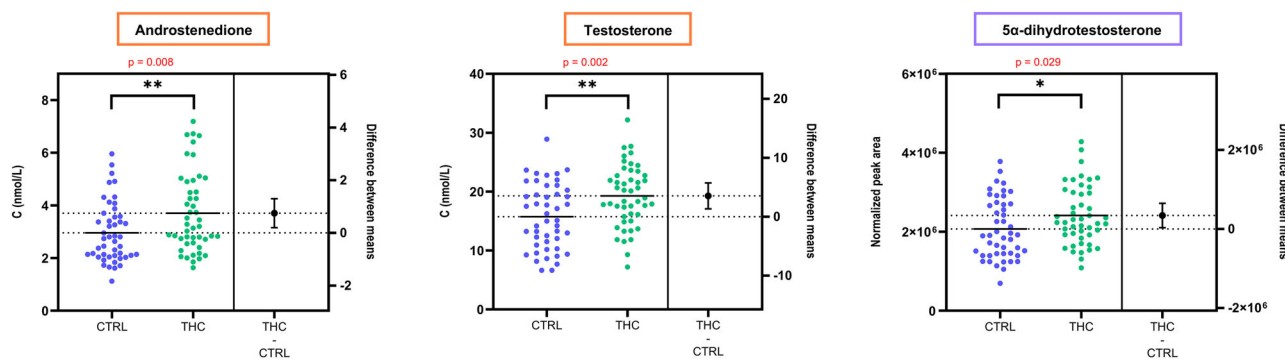

**Fig. 1 | Beeswarm charts with univariate statistical analyses (two-tailed t-tests) showing statistically significant differences in the concentration of the three main bioactive androgens between cannabis users ($n = 47$, green) and controls ($n = 47$, blue).** Error bars indicate the 95% confidence interval (CI) of the difference between group means. Orange squares indicate major steroids that were quantified using one-point internal calibration, whereas purple squares refer to compounds from the extended steroid profile with no absolute quantification data. The right panel of each chart represents the difference between the "THC-positive" and the "Control" group.

observed $Q^2$, meaning that the OPLS-DA model can be considered statistically significant (see Supplementary Fig. S6)[34].

Consistent with absolute quantification, the OPLS-DA model confirmed higher levels of T, A4, and DHT in THC-positive individuals (see Fig. 2). Variable Importance in Projection (VIP) scores further highlighted these androgens as key contributors to the group separation, with T and DHT ranking second and fourth among all variables (see Supplementary Fig. S5).

However, not all androgens contributed equally to the discrimination between cannabis users and non-users. Dehydroepiandrosterone (DHEA), a precursor of androgen synthesis, did not contribute to group separation in the model. As, in contrast, 17α-OHP4 is significantly increased in cannabis users (see Table 1), this suggests that the increase of gonadal androgens (A4, T, and DHT) may be mediated by the Δ4 pathway (involving 17α-OHP4 as precursor of A4) instead of the Δ5 pathway (production of A4 via DHEA).

All C11-oxy androgens (11β-hydroxytestosterone (11β-OHT), 11β-hydroxyandrostenedione (11β-OHA4), 11-ketotestosterone (11-KT), and 11-ketoandrostenedione (11-KA4)) also displayed similar levels in both groups according to M1 loadings (see Fig. 2). Notably, 11β-hydroxydihydrotestosterone (11β-OHDHT) was not among the target compounds, and 11-ketodihydrotestosterone (11-KDHT) was targeted but not detected. These findings were confirmed by univariate analyses, which showed no significant differences in C11-oxy androgen levels between users and non-users ($p > 0.5$, see Supplementary Fig. S7).

C11-oxy androgens are synthesized almost exclusively in the adrenal cortex via the enzymatic activity of the 11β-monooxygenase CYP11B1 and the 11β-dehydrogenase HSD11B2, which act on the precursor androgen substrates[35–38]. As such, their circulating concentrations predominantly reflect adrenal androgenic output[39], in contrast to A4, T, and DHT levels for which testicular synthesis largely prevails in men[33].

It can thus be concluded that adrenal and peripheral androgen biosynthesis in men is not affected by cannabis usage. It is particularly relevant to pinpoint that the highly potent 11-KT, which has similar bioactivity to DHT[36], is not higher in cannabis users, and therefore does not affect androgenic activity compared to non-users.

**Two metabolites of progesterone are strongly related to cannabis consumption**
In the OPLS-DA model M1, the highest and third-highest VIP values were attributed to two progestogens, namely 5β-dihydroprogesterone (5β-DHP4) and 11β-hydroxyprogesterone (11β-OHP4), which were more concentrated in serum from cannabis consumers (see Fig. 2 and Supplementary Fig. S5). Multivariate trends were confirmed by t-tests that revealed a strong relationship between these two steroid compounds and cannabis usage. The high statistical significance of this finding was characterized by a $p$-value of $1.9 \cdot 10^{-5}$ for 5β-DHP4 and $5.7 \cdot 10^{-5}$ for 11β-OHP4 (see Fig. 3A).

5β-DHP4 and 11β-OHP4 are downstream metabolites of progesterone. 5β-DHP4 is formed through the 5β-reductase pathway. Despite its clear association with cannabis use, the functional role of this metabolite in male reproductive biology remains unknown, warranting further biochemical investigation. However, its presence among testicular steroids was recently suggested in a murine gonadal cell model (MA-10), although not delineated from its 5α-counterpart, 5α-dihydroprogesterone[40].

11β-OHP4 is synthesized via the steroid 11β-monooxygenase CYP11B1 and the aldosterone synthase CYP11B2[41]. Interestingly, other CYP11B-derived metabolites, such as 11β-hydroxyandrostenedione (11β-OHA4), did not follow the same trend, suggesting that the increase in 11β-OHP4 is metabolite-specific and not due to generalized upregulation of CYP11B enzymes. In previous reports, 11β-OHP4 was described as a potential precursor of the backdoor pathway and the biosynthesis of C11-oxy steroids[41]. In the present study, downstream metabolites of 11β-OHP4 in this backdoor pathway, such as 11-ketoprogesterone (11-KP4) and 11-ketodihydrotestosterone (11-KDHT), were targeted, but they could not be detected. This precludes further conclusions on the metabolic fate of 11β-OHP4 in the context of cannabis use. However, similarly to 5β-DHP4, this molecule was evidenced in a Leydig cell mouse model, suggesting that it may be directly produced in the testes[40], even though the expression of CYP11B1 in human testes remains uncertain[39,42].

**Phytocannabinoid levels reveal dose-dependent associations with steroidome changes**
A deeper investigation of the relations between androgen levels and THC exposure was performed using Partial Least Squares (PLS) multivariate regression for the THC-positive group, thanks to additional data on serum concentrations of THC and its primary metabolite THC-COOH[21]. THC and THC-COOH concentrations were considered as independent variables, and the 70 variables of the extended steroid profile dataset were used as X-independent variables (THC-model M2 in Supplementary Section S4; THC-COOH model M3 in Supplementary Section S5). Both models displayed similar relationships between cannabis markers and steroid compounds. In particular, the three main gonadal androgens (A4, DHT, and T) systematically increased when THC and THC-COOH levels were higher, as their coefficients in M2 and M3 were all positive (see Fig. S10 and Fig. S15 in the Supplement). On the other hand, the C11-oxy derivatives were either not significantly related to THC and THC-COOH levels (VIP < 1, see Supplementary Figs. S11 and S16) or were negatively correlated with THC and THC-COOH when their contribution to models M2 and M3 was significant (see Supplementary Figs. S10 and S15).

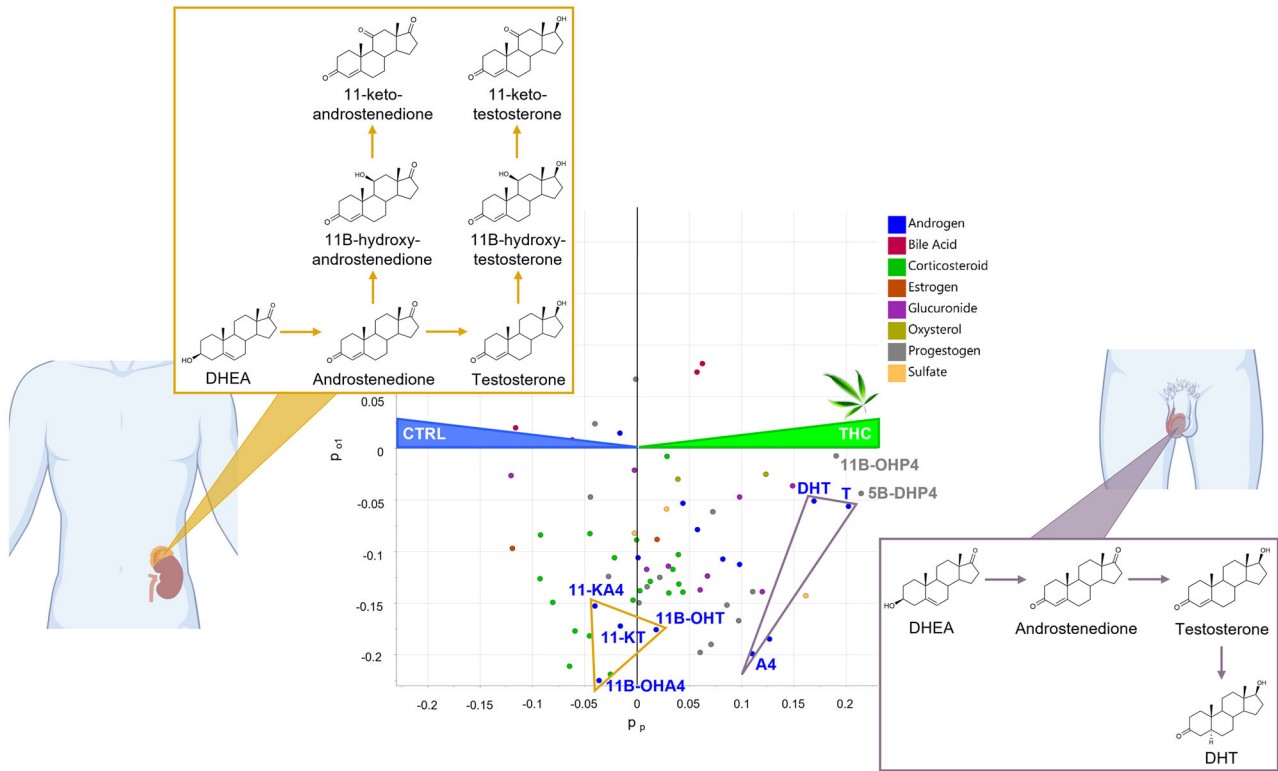

**Fig. 2 | Loadings of OPLS-DA model M1, characterizing the variables from the extended steroid profile which are discriminating serum samples from THC-positive participants (right-hand side) versus controls (left-hand side).** While DHT, T, and A4 are increased in THC-positive samples, C11-oxy androgens display similar levels in both groups. Icons were created in BioRender: M.G. (2026) https://BioRender.com/vrajkgx.

These results highlight that within cannabis users, increased exposure to phytocannabinoids, characterized by higher levels of THC and THC-COOH, is associated with higher levels of testicular androgens, but not with higher levels of C11-oxy androgens (Supplementary Fig. S12). This reinforces the hypothesis of a specific alteration of gonadal—and not adrenal—steroidogenesis upon cannabis exposure.

Cannabis consumers were further stratified into "chronic" ($n = 33$) or "occasional" ($n = 14$) users, based on their blood THC-COOH concentration, a validated biomarker of exposure, using thresholds established by Fabritius et al.[21,22].

In contrast to A4, no significant difference in serum T levels was found between chronic and occasional cannabis consumers ($p = 0.39$, Supplementary Data 4, Supplementary Fig. S17). This is consistent with findings of Fantus et al., who reported a non-linear, "inverse U" relationship between frequency of cannabis use and T levels[19], as well as Thistle et al., who stated that the recency of use, rather than frequency, had a greater influence on circulating testosterone concentrations[11]. In the present study, all "THC-positive" participants declared cannabis use in the last 7 days before sampling, and showed detectable levels of THC in blood. They were thus all recent users and could not be classified more finely according to this criterion. Similarly, no extrapolation to long-term effects on past users can be drawn from the present dataset.

An OPLS-DA model was generated to differentiate chronic and occasional users based on their circulating steroid profile (Model M4, Supplementary Section S7). Interestingly, 5β-DHP4 emerged again as the primary contributor of the discrimination between occasional and chronic consumers (see Supplementary Fig. S19). On the other hand, 11β-OHP4 was not a major contributor to the predictive component of this model (VIP < 1, see Supplementary Fig. S20). Univariate analyses confirmed this trend (see Fig. 3B). The corresponding t-tests showed non-significant differences of 11β-OHP4 levels between chronic and occasional users

($p = 0.10$), supporting its role as a general exposure marker, which does not vary significantly depending on the acuteness of cannabis consumption.

In contrast, 5β-DHP4 levels were significantly higher in chronic users ($p = 0.0027$, see Fig. 3B). Therefore, 5β-DHP4 may serve as a dual biomarker, reflecting both cannabis exposure (Fig. 3A) and the intensity or chronicity of use (Fig. 3B). This is further supported by its positive correlation with circulating THC and THC-COOH levels (see Supplementary Sections S4 and S5).

## Steroidomics provides further insight into the uncertain mechanisms of male hormonal response to phytocannabinoid intake

While previous studies have explored associations between cannabis consumption, semen quality parameters, and serum hormone levels[16,17,21], this is the first to apply comprehensive steroid profiling to healthy young men in relation to their cannabis consumption. A major advantage of our cohort is its narrow age range (18–23 years), which limits age-related hormonal variability seen in other studies[13,16]. Other known confounding factors, such as time of day at sampling, Body Mass Index (BMI), or tobacco smoking[17,43–46], were also carefully investigated and found non-significant in this work (see Section S8 of the Supplement, including Figs. S22 and S23). However, the influence of other confounders originating from the participants' lifestyle, including diet, alcohol consumption, sleep patterns, stress, etc., cannot be ruled out, as some of these parameters could not be assessed. Furthermore, the hereby-presented results reflect short-term effects on recent cannabis users for a restricted group of young Swiss men. At this stage, they should not be generalized to women, other age groups, or more diverse populations. The cross-sectional nature of this study and the wide array of factors influencing steroid metabolism result in an inherent risk of false positives. Every possible precaution was taken to minimize this risk, and follow-up work will be conducted to confirm the outcome of the present study.

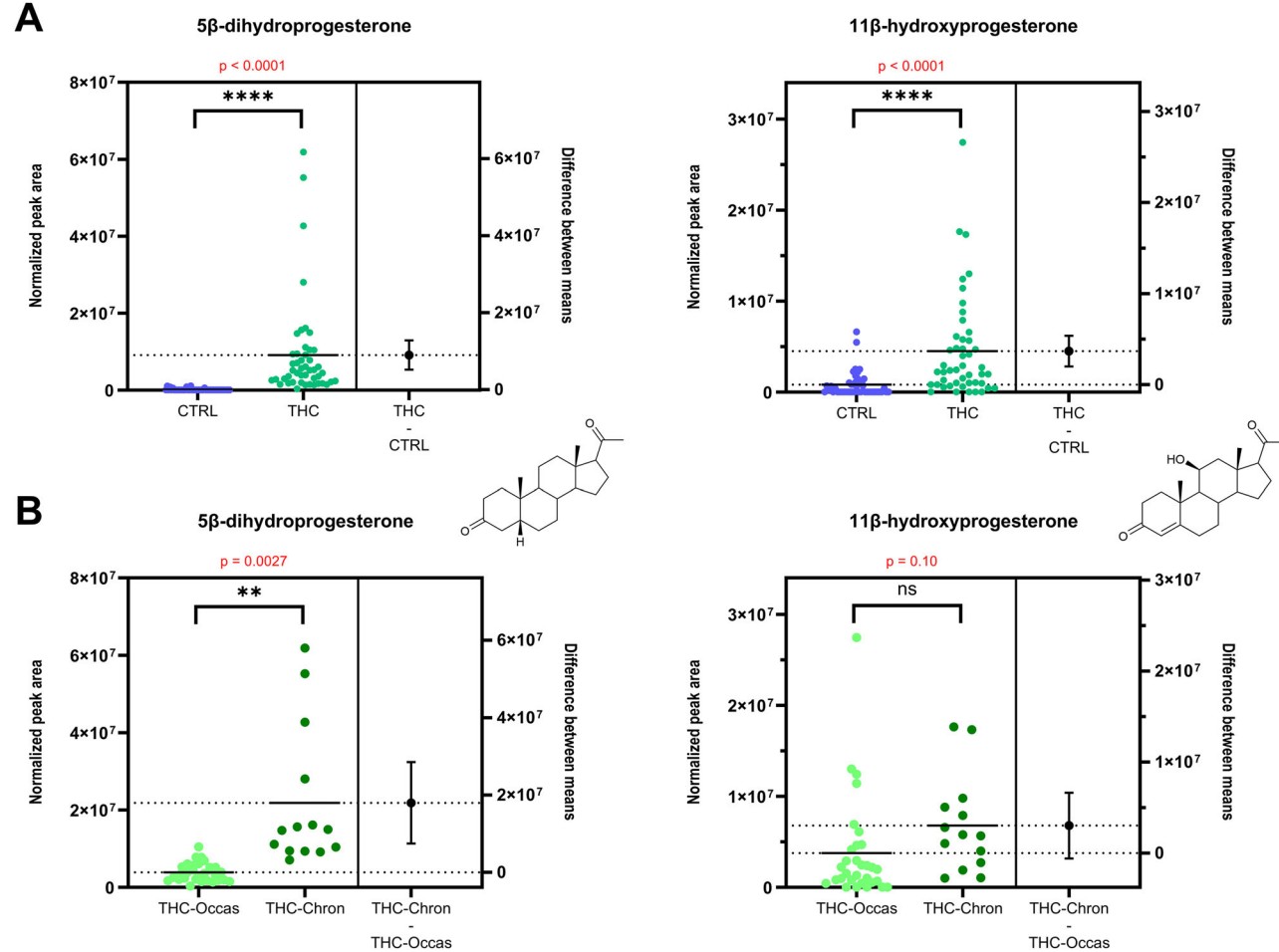

**Fig. 3 | Beeswarm charts with univariate statistical analyses (t-tests) showing statistically significant differences in serum concentration of 5β-DHP4 and 11β-OHP4. A** Between cannabis users (n = 47, green) and non-users (n = 47, blue). **B** Between chronic (n = 14, dark green) and occasional (n = 33, light green) users. Error bars indicate the 95% confidence interval (CI) of the difference between group means. The right panel of each chart represents the difference between the "THC-positive" and the "Control" group (A) or between the chronic and the occasional group (B).

The main finding of this study is the consistent increase of T in cannabis consumers, a trend now corroborated by multiple large-scale cross-sectional studies from both Europe and North America, pointing to higher levels of androgens in cannabis users[17,19,21]. This growing body of evidence challenges earlier reports suggesting testosterone suppression[12,47].

Despite these consistent findings, the underlying mechanisms remain unclear. By combining extended steroid profiling, cannabis biomarker quantification, and broader hormonal analysis, this study enables the exploration of potential pathways responsible for this increase in T.

Fantus et al. proposed several hypotheses to explain elevated T in cannabis users: modulation of LH levels, direct effects on testicular receptors, central suppression, hypothalamic modulation, or interactions at multiple levels. Another possibility is reverse causality, that men with inherently higher testosterone could be more prone to cannabis use due to increased risk-taking behavior[19].

It may indeed be argued that cannabis usage is more likely in men with naturally higher levels of T, as this physiological characteristic could lead to increased risk-taking[21,48,49]. However, the specific relationship between baseline T levels and cigarette smoking or drug abuse was previously described as "significant, but modest"[48], and the relationship between T and risk-taking in general might be overstated due to underestimation of social status as a confounding factor[50].

A more compelling hypothesis is that phytocannabinoids directly disrupt the homeostasis of the HPG axis due to the presence of cannabinoid receptors along this axis. However, in the present study, no relation could be established between LH and/or FSH levels and sex steroid levels (see Supplementary Section S9). Moreover, LH and FSH concentrations did not significantly differ between THC-positive and THC-negative participants (see Supplementary Fig. S24). The complexity of feedback mechanisms, whereby gonadal steroids modulate pituitary hormone secretion, along with the concurrent elevation of LH and gonadal androgens in users, makes it difficult to attribute increased testosterone specifically to LH modulation[38]. An additional limitation of this analysis is the pulsatile secretion of LH and FSH, which reduces the comparability of single-time-point measurements. Likewise, hypothalamic modulation via GnRH cannot be directly evaluated, as GnRH is secreted in pulses and does not circulate in the bloodstream. With no measurable intermediates beyond LH and FSH, evidence for a direct hypothalamic effect of phytocannabinoids remains inconclusive. Thus, while a phytocannabinoid-induced disruption of the HPG axis is a plausible explanation, the precise mechanisms remain unclear. Furthermore, the increase of T may also be seen as a homeostatic response to compensate for a decreased sensitivity of the androgen receptor in the presence of phytocannabinoids.

The present work demonstrates that cannabis-related alterations of androgen metabolism are confined to the gonadal part of the HPG axis. This is evidenced by elevated concentrations of the two other major gonadal androgens, A4 and DHT, while adrenal-derived androgens, such as C11-oxy androgens, remain unaffected. This finding rather supports a possible direct effect of phytocannabinoids on the testicular sex hormone synthesis, mediated by CB1 receptors located in Leydig cells.

Another possible source of steroid metabolism alteration that cannot be ruled out in this study is the effect of phytocannabinoids on liver metabolism. It is known that CB1 and CB2 receptors are expressed in the liver[4,51]. Among the steroids that show increased concentrations upon cannabis exposure according to our multivariate models, several are hydroxylated or reduced metabolites of A4, T, and P4 (see Fig. 2 and Supplementary Figs. S10 and S15), which may indicate increased activity of hepatic cytochrome P450 (CYP) enzymes. However, the literature on the effects of cannabis use on CYP enzymes is conflicting. While both possibilities were reported, there is slightly more in vitro evidence to suggest that THC and CBD inhibit CYP enzymes than induce them, and there is no conclusive evidence from in vivo studies[52]. Therefore, liver metabolism should be considered as a potential factor in cannabis-related alterations of steroid metabolism, but its role in this study remains uncertain and relatively minor compared to testicular steroidogenesis.

Two compounds from the extended steroid profile, 5β-DHP4 and 11β-OHP4, have been highlighted in this work as potential biomarkers of phytocannabinoid intake in men. While 5β-DHP4 might serve both as a biomarker of exposure and a biomarker of intensity, 11β-OHP4 would better characterize a generic exposure to phytocannabinoids with less significant dose-dependency. It is especially interesting to observe alterations of progesterone metabolism in cannabis users, as progesterone plays a key role in reproductive processes such as LH receptor expression, intracellular signaling in sperm, chemotaxis, and acrosome reaction[53–56].

Although this study sheds light on the hormonal effects of cannabis use, its implications for male reproductive health remain uncertain. Data on semen quality in cannabis users from the general population are limited, and findings so far are inconclusive. Recent studies either report no significant differences in semen parameters between users and non-users[21], or point to a reduced sperm concentration and total sperm count[17]. Given the consistent observation of elevated gonadal androgen levels in serum from cannabis users, it is now essential to further investigate how this hormonal profile relates to semen quality in the context of cannabis exposure. There is also a need for further development of relevant in vitro models for the toxicological evaluation of endocrine perturbations in the HPG axis (e.g., human Leydig cells), while current OECD Guidelines for the Testing of Chemicals rely on the adreno-carcinoma H295R cell line to assay steroidogenesis[57].

## Data availability
Preprocessed LC-MS data that support the findings of this study are reported in.xlsx format as Supplementary Data 5. The numerical data plotted in Figs. 1 and 3 can be found in Supplementary Data 6 and 7, respectively. Raw data are available from the corresponding author upon request. All personal and biomedical data on the participants are not publicly available because of privacy or ethical restrictions.

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

## Acknowledgements

The authors would like to thank Oriane Strassel for her help with data pre-processing and absolute quantification, and Marie-Anaïs Monat for her technical support during sample preparation. This work was supported by a grant from the Swiss Centre for Applied Human Toxicology (SCAHT). The collection of human biological material used for this study was supported by the FABER Foundation, the Fondation des Hôpitaux Universitaires de Genève, the Swiss National Science Foundation (SNSF)—NRP 50 "Endocrine Disruptors: Relevance to Humans, Animals and Ecosystems", the Medical Services of the Swiss Army (DDPS), and Medisupport.

## Author contributions

M.G., I.M., J.B., and S.R. designed the experiment. M.G. performed sample preparation, LC-MS data acquisition, and LC-MS data processing. F.Z. performed quantitative analyses of cannabinoids and hormones. R.R., A.S., and S.N. were responsible for sample collection and storage. M.F.R., S.N., and S.R. acquired the funding and managed the collaborative project. M.G. prepared the first draft of the manuscript. M.G., I.M., F.Z., M.F.R., R.R., A.S., S.N., J.B., and S.R. all reviewed and edited the manuscript.

## Competing interests

The authors declare no competing interests.
