## [Transparent Peer Review file · Communications Medicine]

Cannabis consumption is associated with altered steroid metabolism in young men

Corresponding Author: Professor Serge Rudaz

Version 0:

Reviewer comments:

Reviewer #1

(Remarks to the Author)
Comments to the authors

The manuscript describes the alterations of steroid profiles in serum of cannabis users using LC-MS/MS. The main alterations were reported to be increases in androstenedione, testosterone, dihydrotestosterone, 11beta-hydroxyprogesterone, and 5beta-dihydroprogesterone.

While the effects of cannabis consumption on the male reproductive system have been investigated, the findings in the prior studies have been inconsistent. Interestingly, those studies have limitations such as "reliance on self-reported cannabis use without biomarker confirmation", which may have severely influenced the results leading to the inconsistency. In this manuscript, however, the authors quantitatively analyzed the concentrations of THC and THC-COOH to ensure the genuine correlation between the cannabis intake and hormonal levels. An additional advantage of the manuscript is that the authors targeted not just testosterone (the sole target in most studies) but also 170 other steroids to obtain a more complete understanding of the hormonal change. Furthermore, the narrow age range of the cohort minimizes age-related differences. As such, the manuscript would be of interest to those in the field.

The manuscript is very well-written and easy to read. This reviewer has only one comment for the authors. In the manuscript, the authors have compared the differences between chronic and occasional users to evaluate the effects of frequency. However, it seems possible that the recency of use can be a confounding factor. Have the authors evaluated the data in terms of recency as well? It would be beneficial if the authors could provide data or comments concerning recency.

Specific comments

1. P6L150: Table S1 (or any other supplementary tables) could not be found in the submitted documents as far as the reviewer tried. Please provide supplementary tables.
 2. P14L378: It appears that the source of THC and THC-COOH standards in the Methods section is missing.
- Supplementary information
3. P17L192: Figure S38 should read Figure S22.
 4. P18L210: Figure S1 should read Figure S23.

Reviewer #2

(Remarks to the Author)
Reviewer's comments

In the submitted manuscript, Galmiche et al., presents a manuscript detailing the effect of cannabis use on steroid metabolism in a Swiss cohort of young men. State-of-the-art LC-MS/MS was used for the identification and quantification of the circulatory steroid metabolome. Statistical models were applied to analyse differences between cannabis users and matched controls, as well as chronic versus occasional cannabis users. Care was given to the correct statistical methods, together with assessing the possible confounding influence of tobacco smoking and BMI on the results. The influence of cannabis use on gonadal rather than adrenal steroidogenesis, supported by gonadal- and adrenal-specific androgens, is presented and provides an interesting avenue for future investigations relating to male reproductive health and cannabis use.

The paper is well written and follows a good structure. Moreover, this manuscript is a good follow-up to the authors' protocol, supporting the application of their work. Some discussion points, however, need to be added, while some abbreviations need to be defined and the inconsistencies of steroid nomenclature and abbreviations checked. Missing tables also need to be addressed.

Main points which require to be addressed:

1. The authors delineate the influence of cannabis use on male steroid metabolism into gonadal and adrenal steroidogenesis. These two avenues are supported by the data, however, the role of liver steroid metabolism should also be mentioned. Specifically, if Figures S14, S19 and S20 is considered comparing chronic cannabis use with occasional use (and serum THC-COOH levels). These figures highlight 5 β -DHP4 (AKR1D1 catalyses 5 β -DHP4 from P4), 20 α OHP4 (AKR1C1 catalyses 20 α OHP4 from P4), Pdiol-G (the 5 β ,3 α ,G-metabolite of P4), and hydroxylated DHEAS, A4, T and P4 forms (OHDHEAS, 16 α OH-A4, 6 α OH-T, and 11 β OHP4) catalysed by hepatic cytochrome P450 enzymes. Could it be that cannabis use influences liver enzymatic expression or hepatic receptor activity?
2. On p. 9, line 236, the authors refer to CAH (21OHD) patients, which suffer from a genetic block in a steroid pathway due to an enzymatic defect. This block leads to an overflow into other pathways including adrenal androgens and progesterone and its downstream metabolites. Please clarify the text here, that in the referenced study the cohort had an adrenal disorder.
3. Methods (p.15, line 413 & line 424 & line 431): Table S1, S4 and S2 does not seem to exist. Please also double-check as Table S1 and S2 are also referenced on p. 6, line 150 and line 151. In the supplemental material (p. 16, line 168), Table S5 is referenced but not available. Table S3 is mentioned on p.10, line 266 but not shown.

Minor comments:

1. Abstract (p.2, line 38): The authors could consider to write out the abbreviation, LC-MS/MS.
2. Steroid nomenclature: Please use α/β to identify the configurations consistently.
3. Abbreviations (p.4, line 109): Please write out the abbreviation for DHEAS upon its first mention in the introduction. Please check the manuscript for abbreviation inconsistencies.
4. Methods (p. 14, lines 375-376): Please define if spot urine or 24h-urine was used for the quantification of THC-COOH levels in urine. 5. Please also provide the cut-off values which distinguish chronic from occasional users, together with the concentration level in serum which defines cannabis users. If not here, then please add this detail in text on p. 10, line 264.
6. Results & Discussion (p.5, lines 133 - 135): Please abbreviate the steroids. See also p. 7, line 181 (and throughout the manuscript).
7. Supplementary file (p. 17, line 192): Figure S38 should be Figure S22.
8. Supplementary file (p. 18, line 210): Please double check if the reference here to Figure S1 is correct.

Reviewer #3

(Remarks to the Author)

This relevant study reports the effects of regular THC containing cannabis use on endocrine function by performing an extended steroid profiling in cannabis consumers and matched controls. Using a highly sensitive LC-MS/MS platform, the researchers reliably identified 70 endogenous steroids in serum, with seven major steroids subjected to quantification. Multivariate analyses revealed a general increase in androgen levels among cannabis users, with androstenedione (A4), testosterone (T), and dihydrotestosterone (DHT) showing the most significant elevations. In contrast, C11-oxy androgens remained unchanged, suggesting that cannabis may selectively influence gonadal androgen synthesis rather than adrenal or peripheral steroid pathways—potentially through direct testicular effects or disruption of the hypothalamic–pituitary–gonadal (HPG) axis. Two progesterone metabolites, 11 β -hydroxyprogesterone (11B-OHP4) and 5 β -dihydroprogesterone (5B-DHP4), were also significantly elevated, with 11B-OHP4 identified as a general exposure biomarker and 5B-DHP4 showing an intriguing dose-dependent relationship to cannabis use.

Overall, the study is methodologically robust, with some questions remaining (see below) and provides novel insights into how cannabis consumption may modulate male endocrine function through specific steroid pathways. Potential issues: The study's conclusions should be interpreted more cautiously in light of sample size, multiplicity, and cross-sectional design (see comments below).

Comments:

In Fig. 1, androstenedione and possibly others seem to have a group of 7 participants that show higher values driving the significance in the analysis. The authors state "All samples were taken in late afternoon, considerably reducing the influence of diurnal variations as confounder". Hormones such as testosterone, cortisol, androstenedione, and progesterone metabolites fluctuate substantially across the 24-hour cycle — often peaking in the early morning and declining during the day. For example, testosterone can vary by 30–50% depending on the time of blood draw. If cannabis users and controls were not sampled at the same circadian phase and also regarding differences in sleep/wake schedules due to lifestyle, the observed "global increase in androgen levels" could be partly influenced by differences in sampling time or disruption of circadian regulation. Action: The authors should therefore provide a table of the sampling time, or show the lack of correlation of time between groups. Was the sleep/wake schedules similar between the groups?

Fig. 1 and 3: The THC control should be explained in the legend. It is not immediately clear what this control is exactly.

Maybe the most meaningful figures in the suppl. showing the THC and THC-COOH level correlations with steroids could be included in the main manuscript, for clarity.

Methods:

A major strength is that cannabis exposure was objectively verified through serum THC and metabolite measurements rather than self-report. The use of LC-MS/MS with internal standards ensured high analytical precision, and seven key steroids were quantified. It is somewhat unclear why some were not absolutely quantified (Fig. 3), given the possibility for a MRM method?

However, the cross-sectional design may include causal inference, and the relatively small sample size (47 users and 47 controls) does limit statistical power, particularly given that around 70 steroids were analyzed without an explicit correction for multiple comparisons, increasing the risk of false positives. The OPLS-DA model used to distinguish groups showed limited predictive strength ($Q^2 \approx 0.19$), suggesting that group separation may be modest. Additionally, samples were collected over a long period (2005–2017), and while the authors mention quality control, potential batch or storage effects are not fully addressed, though steroids are very stable in blood over time.

Because gonadotropins such as LH and FSH are pulsatile, single time-point afternoon measurements make it difficult to draw firm conclusions about hypothalamic–pituitary–gonadal (HPG) axis activity.

Residual confounding factors such as sleep patterns, chronotype, diet, stress, or co-use of other substances also remain possible, as they were not thoroughly controlled. Can the authors add more details on this? Furthermore, only seven steroids were quantified absolutely, while the rest relied on semi-quantitative normalization that may introduce ionization biases. Finally, the study population—young Swiss men, many recruited through military processes—limits generalizability to other age groups, women, or more diverse populations.

Replacing missing values with one-third of the mean blank area is somewhat arbitrary. Common alternatives may include imputation using e.g. half of the minimum detected value per analyte, KNN or random-forest imputation, or censoring below LOD.

Version 1:

Reviewer comments:

Reviewer #1

(Remarks to the Author)

My concerns have been addressed. I thank the authors for the revision.

Reviewer #2

(Remarks to the Author)

In the resubmitted manuscript by Galmiche et al., all comments have been addressed and appropriate corrections applied to the manuscript. A few minor points, however, still remain to be addressed.

1. The reviewer refers the authors to p.13, lines 350 - 360. For absolute correctness, just a short comment: in the referenced article 5α -DHP4 and 5β -DHP4 were not chromatographically separated and therefore both metabolites were quantified as one. This does not delineate the 5α - from the 5β -reduction pathway in the murine Leydig cell line (unfortunately). Additionally, the quantification of 11β -hydroxyprogesterone in the MA-10 cell line does not directly relate to 11β -hydroxylase expression in human testes as evidenced in the following papers: <https://doi.org/10.1210/clinem/dgae420> & DOI: 10.1530/EJE-22-0518 (ovary data). A previous publication has, however, shown the presence of 11 -ketotestosterone (dependent on the prerequisite that 11β -hydroxytestosterone is produced) and the expression of CYP11B1 in human Leydig cells: DOI: 10.1210/jc.2016-2311. The following supportive data in rat Leydig cells: DOI: 10.1210/endo.143.2.8638. Please consider adding these last two mentioned additional references to support the statement 'suggesting the possibility of its direct testicular synthesis'.
2. Table S1 is still missing. Is Table S1 perhaps on p.17 of the supplementary file (previously implied as Table S5)? Please correct numbering.
3. Please abbreviate testosterone as T upon first mention in the introduction (p. 4) and androstenedione as A4 upon first mention in the introduction (p. 5). Please add these abbreviations also on p. 9 (line 257), 16 (line 432) and 17 (line 470) in the manuscript.
4. Please abbreviate water/methanol and formic acid on p.7 (lines 195-197), as these were abbreviated previously on p. 6.

Reviewer #3

(Remarks to the Author)

The authors have addressed all the critical points in their review and answered satisfactorily all questions in their rebuttal. I would like to thank the authors for their replies and for significantly improving their manuscript. I believe that the manuscript is acceptable for publication.

Referee expertise:

Referee #1: Metabolomic profiling, cannabinoids

Referee #2: Mass spectrometry, steroid analyses, hormone physiology

Referee #3: Mass spectrometry, cannabinoids

Reviewers' comments:**Reviewer #1 (Remarks to the Author):**

Comments to the authors

The manuscript describes the alterations of steroid profiles in serum of cannabis users using LC-MS/MS. The main alterations were reported to be increases in androstenedione, testosterone, dihydrotestosterone, 11beta-hydroxyprogesterone, and 5beta-dihydroprogesteron

While the effects of cannabis consumption on the male reproductive have been investigated, the findings in the prior studies have been inconsistent. Interestingly, those studies have limitations such as “reliance on self-reported cannabis use without confirmation”, which may have severely influenced the results leading to the inconsistency. In this manuscript, however, the authors quantitatively analyzed the concentrations of THC and THC-COOH to ensure the genuine correlation between the cannabis intake and hormonal levels. An additional advantage of the manuscript is that the authors targeted not just testosterone (the sole target in most studies) but also 170 other steroids to obtain a more complete understanding of the hormonal chang Furthermore, the narrow age range of the cohort minimizes age-related differences. As such, the manuscript would be of interest to those in the field.

We thank Reviewer #1 for his/her positive review of our manuscript.

The manuscript is very well-written and easy to read. This reviewer has only one comment for the authors. In the manuscript, the authors have compared the differences between chronic and occasional users to evaluate the effects of frequency. However, it seems possible that the recency of use can be a confounding factor. Have the authors evaluated the data in terms of recency as well? It would be beneficial if the authors could provide data or comments concerning recency.

We specifically asked participants in the questionnaires whether they had consumed marijuana during the past week. Moreover, all participants involved in the present study showed detectable levels of THC in blood, demonstrating very recent consumption. Therefore, all reported values already reflect very recent use. Recency is inherently built into the study, and no additional temporal categorization (e.g., past month, lifetime use) was collected.

In this revised version of the manuscript, we now clarify this point in the Methods section (page 6). Substance use refers to use within the previous seven days, ensuring that the exposure reflects recent behavior. We also added a comment in the Discussion noting that our data capture short-term recency, but do not allow evaluation of longer-term patterns of use (page 14).

The only possibility to evaluate further the effect of recency of use was to apply models from Huestis et al. (<https://doi.org/10.1093/jat/16.5.283>), which estimate the time since last cannabis consumption based on blood concentrations of THC and THC-COOH. However, these models are known to be imperfect, especially for infrequent users. In our study, the estimated times since last use were below 12 hours for all 47 participants, with a mean value around 3 hours. Based on these data, no PLS model could efficiently relate the estimated time since last use and the steroid profile.

Specific comments

1. P6L150: Table S1 (or any other supplementary tables) could not be found in the submitted documents as far as the reviewer tried. Please provide supplementary tables.

We apologize for the absence of the Supplementary Tables in the first version of the manuscript. This is now solved in the revised submission.

2. P14L378: It appears that the source of THC and THC-COOH standards in the Methods section is missing.

This is because the quantification of THC and THC-COOH was performed in a previous study (DOI:10.1111/andr.13440), as it is now reported in the Methods section (page 6). The resulting data were combined with the steroidomics data from the present study. However, to comply with the comment of Reviewer #1, we reported the analytical procedure for THC and THC-COOH from this previous study in the revised Supplementary Information (Section S10).

Supplementary information

3. P17L192: Figure S38 should read Figure S22.

4. P18L210: Figure S1 should read Figure S23.

Indeed, both were typos. We thank Reviewer #1 for bringing this to our attention, and we have now corrected the issue.

Reviewer #2 (Remarks to the Author):

Reviewer's comments

In the submitted manuscript, Galmiche et al., presents a manuscript detailing the effect of cannabis use on steroid metabolism in a Swiss cohort of young men. State-of-the-art LC-MS/MS was used for the identification and quantification of the circulatory steroid metabolom. Statistical models were applied to analyse differences between cannabis users and matched controls, as well as chronic versus occasional cannabis users. Care was given to the correct statistical methods, together with assessing the possible confounding influence of tobacco smoking and BMI on the results. The influence of cannabis use on gonadal rather than adrenal steroidogenesis, supported by gonadal- and adrenal-specific androgens, is presented and provides an interesting avenue for future investigations relating to male reproductive health and cannabis us

The paper is well written and follows a good structur. Moreover, this manuscript is a good follow-up to the authors' protocol, supporting the application of their wor. Some discussion points, however, need to be added, while some abbreviations need to be defined and the inconsistencies of steroid nomenclature and abbreviations checked. Missing tables also need to be addressed

We thank Reviewer #2 for his/her positive review of our manuscript.

Main points which require to be addressed:

1. The authors delineate the influence of cannabis use on male steroid metabolism into gonadal and adrenal steroidogenesis. These two avenues are supported by the data, however, the role of liver steroid metabolism should also be mentioned. Specifically, if Figures S14, S19 and S20 is considered comparing chronic cannabis use with occasional use (and serum THC-COOH levels). These figures highlight 5 β -DHP4 (AKR1D1 catalyses 5 β -DHP4 from P4), 20 α OHP4 (AKR1C1 catalyses 20 α OHP4 from P4), Pdiol-G (the 5 β ,3 α ,G-metabolite of P4), and hydroxylated DHEAS, A4, T and P4 forms (OHDHEAS, 16 α OH-A4, 6 α OH-T, and 11 β OHP4) catalysed by hepatic cytochrome P450 enzymes. Could it be that cannabis use influences liver enzymatic expression or hepatic receptor activity?

We thank Reviewer #2 for pointing this out. This is a sound hypothesis, as cannabinoid receptors (CB1 and CB2) are also expressed in the liver. A thorough review of the literature revealed that THC and CBD were most frequently reported as CYP inhibitors, and only in very rare cases as CYP inducers.

(doi.org/10.1080/03602532.2024.2346767 ; doi.org/10.2174/1389200217666151210142051 ; doi.org/10.1124/dmd.121.000734).

The state of the art would thus rather advocate against the hypothesis of increased formation of downstream metabolites of testosterone and progesterone in the liver by CYP enzymes. Furthermore, no consistent trend was observed with respect to the implicated enzymes. As Reviewer #2 mentions, several distinct enzymes are involved in the synthesis of these oxygenated or reduced metabolites, and none of them shows a systemic trend on several different substrates. Finally, the production of reduced and oxygenated metabolites of progesterone was also demonstrated in cell cultures such as murine Leydig cells, supporting that the liver is not the sole organ involved in the synthesis of these steroid metabolites (doi.org/10.3390/ijms26199721).

To summarize, although a liver metabolism hypothesis was included in the revised version of the manuscript, we consider that a focus on gonadal and adrenal steroidogenesis is more strongly supported by our data. As requested, a corresponding paragraph was added in the revised manuscript, page 17.

2. On p. 9, line 236, the authors refer to CAH (21OHD) patients, which suffer from a genetic block in a steroid pathway due to an enzymatic defect. This block leads to an overflow into other pathways including adrenal androgens and progesterone and its downstream metabolites. Please clarify the text here, that in the referenced study the cohort had an adrenal disorder.

We understand that the reference that was cited here to provide some literature background may not be appropriate, as it refers to CAH patients, not to healthy men. Therefore, we chose to delete this statement and replaced it with evidence of the detection of this metabolite in murine Leydig cell culture samples (page 13).

3. Methods (p.15, line 413 & line 424 & line 431): Table S1, S4 and S2 does not seem to exist. Please also double-check as Table S1 and S2 are also referenced on p. 6, line 150 and line 151. In the supplemental material (p. 16, line 168), Table S5 is referenced but not available Table S3 is mentioned on p.10, line 266 but not shown.

We apologize for the absence of the Supplementary Tables in the first version of the manuscript. This is now solved in the revised submission.

Minor comments:

1. Abstract (p.2, line 38): The authors could consider to write out the abbreviation, LC-MS/MS. The abbreviation is now explicitly defined.
2. Steroid nomenclature: Please use α/β to identify the configurations consistently. The corresponding modifications have been applied in the revised manuscript.
3. Abbreviations (p.4, line 109): Please write out the abbreviation for DHEAS upon its first mention in the introduction. Please check the manuscript for abbreviation inconsistencies. This abbreviation is now explicitly defined.
4. Methods (p. 14, lines 375-376): Please define if spot urine or 24h-urine was used for the quantification of THC-COOH levels in urine. Although THC-COOH was quantified in spot urine for other purposes in this cohort, these data were not used in the present study. Classification of occasional and chronic cannabis users was actually based on THC-COOH concentrations in serum. This has been corrected in the revised manuscript.
5. Please also provide the cut-off values which distinguish chronic from occasional users, together with the concentration level in serum which defines cannabis users. If not here, then please add this detail in text on p. 10, line 264. A corresponding precision has been added in the 'Sample selection' section of the revised manuscript (page 6).
6. Results & Discussion (p.5, lines 133 - 135): Please abbreviate the steroids. See also p. 7, line 181 (and throughout the manuscript). Abbreviations were checked and corrected throughout the manuscript.
7. Supplementary file (p. 17, line 192): Figure S38 should be Figure S22. This typo has been corrected in the revised Supplement.
8. Supplementary file (p. 18, line 210): Please double check if the reference here to Figure S1 is correct. This typo has been corrected in the revised Supplement.

Reviewer #3 (Remarks to the Author):

This relevant study reports the effects of regular THC containing cannabis use on endocrine function by performing an extended steroid profiling in cannabis consumers and matched controls. Using a highly sensitive LC-MS/MS platform, the researchers reliably identified 70 endogenous steroids in serum, with seven major steroids subjected to quantification. Multivariate analyses revealed a general increase in androgen levels among cannabis users, with androstenedione (A4), testosterone (T), and dihydrotestosterone (DHT) showing the most significant elevations. In contrast, C11-oxy androgens remained unchanged, suggesting that cannabis may selectively influence gonadal androgen synthesis rather than adrenal or peripheral steroid pathways—potentially through direct testicular effects or disruption of the hypothalamic–pituitary–gonadal (HPG) axis. Two progesterone metabolites, 11 β -hydroxyprogesterone (11B-OHP4) and 5 β -dihydroprogesterone (5B-DHP4), were also significantly elevated, with 11B-OHP4 identified as a general exposure and 5B-DHP4 showing an intriguing dose-dependent relationship to cannabis use.

Overall, the study is methodologically robust, with some questions remaining (see below) and provides novel insights into how cannabis consumption may modulate male endocrine function through specific steroid pathways. Potential issues: The study's conclusions should be interpreted more cautiously in light of sample size, multiplicity, and cross-sectional design (see comments below).

We thank Reviewer #3 for his/her positive review of our manuscript.

Comments:

In Fig. 1, androstenedione and possibly others seem to have a group of 7 participants that show higher values driving the significance in the analysis. The authors state "All samples were taken in late afternoon, considerably reducing the influence of diurnal variations as confounder". Hormones such as testosterone, cortisol, androstenedione, and progesterone metabolites fluctuate substantially across the 24-hour cycle — often peaking in the early morning and declining during the day. For example, testosterone can vary by 30–50% depending on the time of blood draw. If cannabis users and controls were not sampled at the same circadian phase and also regarding differences in sleep/wake schedules due to lifestyle, the observed "global increase in androgen levels" could be partly influenced by differences in sampling time or disruption of circadian regulation. Action: The authors should therefore provide a table of the sampling time, or show the lack of correlation of time between groups. Was the sleep/wake schedules similar between the groups?

We thank Reviewer #3 for bringing this potential confounder to our attention. To reply as comprehensively as possible to his/her concerns, we added a new subsection in the revised version of the Supplementary Information (Section S8.a) and sampling time is now mentioned in the manuscript as a potential confounder (Page 15).

Briefly, we managed to retrieve the exact sampling times for each participant and report this data as Figure S22. No significant difference between the two groups was observed.

We also want to emphasize that there was no significant difference in cortisol levels between the groups ($p = 0.72$, see Table 1), whereas cortisol would be the most susceptible to circadian differences between the samples according to the literature (doi.org/10.1016/j.jpsychires.2010.04.015).

We particularly checked the sampling time of outlier values for androstenedione and testosterone, as suggested by Reviewer #3. All corresponding blood samples were taken between 3 pm and 5 pm, i.e. close to the average sampling time in this study. They do not stem from differences in time of blood draw.

However, we agree that variations in steroid concentrations due to individual habits and circadian rhythm cannot be completely ruled out. At recruitment, sleep/wake schedules were not assessed. However, as the majority of samples were obtained in the late afternoon, a non-significant influence of minor shifts in wake-up time on steroid concentrations is expected.

According to the extensive review of Collomp et al. on this topic (dx.doi.org/10.1016/j.physbeh.2016.05.039), only cortisol levels would show an awakening effect within the first hour of waking. This pattern is not expected for androgens.

Fig. 1 and 3: The THC control should be explained in the legend. It is not immediately clear what this control is exactly.

If Reviewer #3 refers to the right panel of Figures 1 and 3, 'THC – CTRL' is the difference between the mean of the 'THC-positive' group and the mean of the control group. We clarified this point directly in the figure captions in the revised manuscript.

Maybe the most meaningful figures in the suppl. showing the THC and THC-COOH level correlations with steroids could be included in the main manuscript, for clarity.

We do not think that adding figures will improve the clarity of the manuscript, but rather the opposite. The restriction of the main manuscript content to the essentials contributes to the ease of reading, which was appreciated by the Reviewers in this evaluation.

Methods:

A major strength is that cannabis exposure was objectively verified through serum THC and metabolite measurements rather than self-report. The use of LC-MS/MS with internal standards ensured high analytical precision, and seven key steroids were quantified. It is somewhat unclear why some were not absolutely quantified (Fig. 3), given the possibility for a MRM method?

This dual analytical strategy featuring both absolute quantification for seven major steroids and an extended profiling of all other steroid metabolites stems from a previously published protocol (doi.org/10.1002/jssc.70147). Due to the challenges inherent to the quantification of endogenous analytes, our approach relies on a one-point internal calibration strategy that is only possible when ¹³C-labeled standards are commercially available for the corresponding analyte, which is the case only for a limited number of steroids.

We understand that this was not clear in the submitted manuscript, and we have now clarified this issue in the corresponding section (page 7).

However, the cross-sectional design may include causal inference, and the relatively small sample size (47 users and 47 controls) does limit statistical power, particularly given that around 70 steroids were analyzed without an explicit correction for multiple comparisons, increasing the risk of false positives. The OPLS-DA used to distinguish groups showed limited predictive strength ($Q^2 \approx 0.19$), suggesting that group separation may be modest. Additionally, samples were collected over a long period (2005–2017), and while the authors mention quality control, potential batch or storage effects are not fully addressed, though steroids are very stable in blood over time.

This cross-sectional study was designed to observe differences in steroidome related to cannabis consumption. All possible precautions were taken to minimize biases. Serum samples were stored at -80°C to minimize possible degradation. This is now clarified on page 7. As steroids are indeed known to be very stable, and as samples from various sampling dates were equally present in both groups, this issue can be considered as very minor.

The data analysis strategy aimed to explore the dataset through multivariate models and generate hypotheses. Therefore, only the variables that seemed to be the main drivers of these multivariate models were further studied via univariate analysis. Not all measured steroids were subject to univariate investigation, which is why correction for multiple comparisons was not performed.

The low Q^2 values of OPLS-DA models may indicate a risk of over-fitting, but cross-validation with permutation tests (presented in Figures S6 and S21) still demonstrates a reasonable significance of these models despite their limited predictive ability. Moreover, significant differences were confirmed by two-tailed t-tests for the most relevant steroid metabolites in this study (e.g., 5β -dihydroprogesterone, 11β -hydroxyprogesterone, 5α -dihydrotestosterone, Testosterone).

However, we agree with the mentioned risk of false positives. This study remains exploratory, and follow-up studies will be conducted to confirm these findings (see the added comment in the revised manuscript, page 15-16).

Because gonadotropins such as LH and FSH are pulsatile, single time-point afternoon measurements make it difficult to draw firm conclusions about hypothalamic–pituitary–gonadal (HPG) axis activity.

We agree with this comment, and we have mentioned this limitation in the revised manuscript (page 16-17).

Residual confounding factors such as sleep patterns, chronotype, diet, stress, or co-use of other substances also remain possible, as they were not thoroughly controlled. Can the authors add more details on this? Furthermore, only seven steroids were quantified absolutely, while the rest relied on semi-quantitative normalization that may introduce ionization biases. Finally, the study population—young Swiss men, many recruited through military processes—limits generalizability to other age groups, women, or more diverse populations.

In accordance with this remark, we added some comments on the limitations of the present work in the revised manuscript, page 15.

However, the use of peak areas instead of absolute concentrations does not constitute an issue because unit variance scaling was applied to the dataset before multivariate modeling. The latter is a common practice for exploratory analysis in metabolomics to address differences in concentration and/or ionizability. In most cases, endogenous concentrations of steroid metabolites remain in the same order of magnitude (see Table 1, Max / Min ratios < 10), thus ensuring a linear MRM response in this narrow range.

Replacing missing values with one-third of the mean blank area is somewhat arbitrary. Common alternatives may include imputation using g. half of the minimum detected value per analyte, KNN or random-forest imputation, or censoring below LOD.

We acknowledge that replacing missing values with one-third of the mean blank area is a pragmatic but somewhat arbitrary approach. However, this method is commonly applied in metabolomics studies to avoid “zeros” in the dataset for multivariate analysis without overestimating concentrations, while preserving variability in the dataset.

More sophisticated imputation strategies, such as KNN or RF imputation, rely on strong correlation structures and/or large sample sizes. These criteria were not met in our relatively small and heterogeneous dataset. Additionally, our primary goal was exploratory rather than predictive, making simpler approaches more appropriate.

Reviewers' comments:

Reviewer #1 (Remarks to the Author):

My concerns have been addressed. I thank the authors for the revision.
We thank Reviewer #1 for supporting the publication of our manuscript with his/her constructive feedback.

Reviewer #2 (Remarks to the Author):

In the resubmitted manuscript by Galmiche et al., all comments have been addressed and appropriate corrections applied to the manuscript. A few minor points, however, still remain to be addressed.

We thank Reviewer #2 for the positive evaluation of our manuscript in its revised version. We carefully address his/her further comments as detailed below.

1. The reviewer refers the authors to p.13, lines 350 - 360. For absolute correctness, just a short comment: in the referenced article 5α -DHP4 and 5β -DHP4 were not chromatographically separated and therefore both metabolites were quantified as one. This does not delineate the 5α - from the 5β -reduction pathway in the murine Leydig cell line (unfortunately).

The corresponding precision was added in the new version of the manuscript. Thank you for bringing this to our attention.

Additionally, the quantification of 11β -hydroxyprogesterone in the MA-10 cell line does not directly relate to 11β -hydroxylase expression in human testes as evidenced in the following papers: <https://doi.org/10.1210/clinem/dgae420> & DOI: 10.1530/EJE-22-0518 (ovary data). A previous publication has, however, shown the presence of 11 -ketotestosterone (dependent on the prerequisite that 11β -hydroxytestosterone is produced) and the expression of CYP11B1 in human Leydig cells: DOI: 10.1210/jc.2016-2311. The following supportive data in rat Leydig cells: DOI: 10.1210/endo.143.2.8638. Please consider adding these last two mentioned additional references to support the statement 'suggesting the possibility of its direct testicular synthesis'.

We thank Reviewer #2 for providing us with these insights into steroidogenic enzymes and organs. This point is critical in the present manuscript.

It is however very tricky, as even within the references suggested by Reviewer #2, some conclude against a significant synthesis of 11 -oxygenated androgens in the testes (Charoensri et al., JCEM, 2025), whereas others support this possible synthesis route (Imamichi et al., JCEM, 2016).

Furthermore, our discussion in this part of the manuscript regards 11β -hydroxyprogesterone, which may be produced by CYP11B enzymes in a substrate-specific manner, differently from 11 -oxygenated androgens: "*the increase in 11β -OHP4 is metabolite-specific and not due to generalized upregulation of CYP11B enzymes.*".

Knowing that our data reflect systemic changes in circulating steroid levels, and that they are thus inherently inappropriate for a definitive mechanistic interpretation, we propose in the revised version of the manuscript a discussion on 11β -OHP4 that balances the various hypotheses on CYP11B1 expression in the gonads and that includes the references proposed by Reviewer #2.

2. Table S1 is still missing. Is Table S1 perhaps on p.17 of the supplementary file (previously implied as Table S5)? Please correct numbering.

This issue is probably related to the submission platform. The editorial team from *Communications Medicine* requires the submission of a single 'Supplementary' file as pdf. However, Table S1 is too cumbersome to be readable in this form. Hence its submission, after revision, as 'Supplementary Data' in a separate Excel file, similarly to the LC-MS dataset. Perhaps Reviewer #2 was not able to access it?

3. Please abbreviate testosterone as T upon first mention in the introduction (p. 4) and androstenedione as A4 upon first mention in the introduction (p. 5). Please add these abbreviations also on p. 9 (line 257), 16 (line 432) and 17 (line 470) in the manuscript.

4. Please abbreviate water/methanol and formic acid on p.7 (lines 195-197), as these were abbreviated previously on p. 6.

We thank Reviewer #2 for noticing these inconsistencies. Abbreviations were corrected throughout the manuscript according to these suggestions.

Reviewer #3 (Remarks to the Author):

The authors have addressed all the critical points in their review and answered satisfactorily all questions in their rebuttal. I would like to thank the authors for their replies and for significantly improving their manuscript. I believe that the manuscript is acceptable for publication.

We thank Reviewer #3 for supporting the publication of our manuscript with his/her constructive feedback.